# Axonal and Glial PIEZO1 and PIEZO2 Immunoreactivity in Human Clitoral Krause’s Corpuscles

**DOI:** 10.3390/ijms25126722

**Published:** 2024-06-18

**Authors:** Patricia Cuendias, José A. Vega, Olivia García-Suárez, Iván Suazo, Ramón Cobo, Jorge García-Piqueras, Yolanda García-Mesa

**Affiliations:** 1Grupo de Investigación SINPOS, Departamento de Morfología y Biología Celular, Universidad de Oviedo, 33006 Oviedo, Spain; cuendiaspatricia@uniovi.es (P.C.); javega@uniovi.es (J.A.V.); garciaolivia@uniovi.es (O.G.-S.); jorge.garciap@uam.es (J.G.-P.); 2Facultad de Ciencias de la Salud, Universidad Autónoma de Chile, Providencia, Santiago de Chile 4810010, Chile; ivan.suazo@uautonoma.cl; 3Instituto de Investigación Sanitaria del Principado de Asturias (ISPA), 33011 Oviedo, Spain; 4Servicio de Otorrinolaringología, Hospital Universitario “Marqués de Valdecilla”, 39008 Santander, Spain; ramoncobodiaz@gmail.com; 5Departamento de Anatomía, Histología y Neurociencia, Universidad Autónoma de Madrid, 28049 Madrid, Spain

**Keywords:** Krause’s corpuscles, terminal glial cells, PIEZO mechanoproteins, clitoris, human

## Abstract

Krause’s corpuscles are typical of cutaneous mucous epithelia, like the lip vermillion or the glans clitoridis, and are associated with rapidly adapting low-threshold mechanoreceptors involved in gentle touch or vibration. PIEZO1 and PIEZO2 are transmembrane mechano-gated proteins that form a part of the cationic ion channels required for mechanosensitivity in mammalian cells. They are involved in somatosensitivity, especially in the different qualities of touch, but also in pain and proprioception. In the present study, immunohistochemistry and immunofluorescence were used to analyze the occurrence and cellular location of PIEZO1 and PIEZO2 in human clitoral Krause’s corpuscles. Both PIEZO1 and PIEZO2 were detected in Krause’s corpuscles in both the axon and the terminal glial cells. The presence of PIEZOs in the terminal glial cells of Kraus’s corpuscles is reported here for the first time. Based on the distribution of PIEZO1 and PIEZO2, it may be assumed they could be involved in mechanical stimuli, sexual behavior, and sexual pleasure.

## 1. Introduction

The body surface of mammals contains cutaneous end-organ complexes (CEOCs), or sensory corpuscles, which are the sensory formations responsible for detection of innocuous mechanosensing and touch [1,2]. Structurally, they consist of the peripheral process extreme tip from primary mechanosensory neurons (low-threshold mechanoreceptors; LTMRs) [2,3,4,5,6] and heterogeneous cells (terminal glial cells, endoneurial and perineurial cells) variably arranged [1,7]. Thus, LTMRs supplying CEOCs are a subset of dorsal root ganglia (DRG) [8] and cranial nerve ganglia neurons responsible for mechanical stimulus detection from both the environment and internal organs. Functionally, LTMRs fall into four categories: type I and type II rapidly adapting (RA) LTMRs and type I and type II slowly adapting (SA) LTMRs, which in mammalian glabrous skin form a part of Meissner’s corpuscles, Pacinian corpuscles, Merkel cell–neurite complexes and Ruffini’s corpuscles, respectively [5].

On the other hand, some mammalian organs like the clitoris are covered by cutaneous mucous epithelia and contain a special morphotype of sensory corpuscle-denominated Krause’s corpuscles (also known as genital corpuscles, genital nerve bodies, genital endbulbs, or glomerular corpuscles), responsible for detection of mechanical vibration and sexual mechanical sensing [9]. Recently, it has been demonstrated that Krause’s corpuscles are RA-LTMRs [10].

Mechanotransduction, i.e., the process whereby mechanical stimuli are converted into an action potential, occurs within sensory corpuscles. This process involves ion channels, especially the members from the Piezo cationic ion channel family, PIEZO1 and PIEZO2 [11]. They are multipass transmembrane proteins required for mechanotransduction in most mammalian cells, with homologs in invertebrates, plants, and protozoa [12]. Consistently, PIEZOs are expressed in mechanoreceptors [12,13,14,15], and PIEZO2 is found in the axon of murine and human CEOCs independently of their anatomical location [12,16,17,18,19], including Krause’s corpuscles from the murine and human clitoris [9,10]. Conversely, only one study has reported the occurrence of axonal PIEZO1 in human digital skin Meissner’s corpuscles [16] but not in sensory corpuscles from other anatomical locations. Nevertheless, Shin et al. [14,15] observed the presence of PIEZO1 and PIEZO2 in both the axon and non-neuronal cells of rat CEOCs, while Yamanishi and Iwabuchi [20,21] revealed PIEZO2 expression in both the axon and terminal glial cells of human lanceolate nerve endings. This double cellular localization of mechano-gated protein in the axons and associated terminal glial cells of sensory corpuscles is of great interest because evidence is being accumulated that not only axons but also non-neuronal cells participate in touch detection [22,23].

Last year, Lam and co-workers showed PIEZO2 is necessary for behavioral sensitivity to perineal touch and is needed for triggering the touch-evoked erection reflex and successful mating in both male and female mice. They also observed that humans with complete loss of PIEZO2 function have genital hyposensitivity and experience no direct pleasure from gentle touch or vibration [24]. Consistently, Krause’s corpuscles from both murine penis and clitoris [10] and human clitoral and preputial CEOCs [9,25] express axonal PIEZO2. However, as far as we know, PIEZO1 occurrence in human Krause’s corpuscles has never been investigated. Thus, we used here immunofluorescence visualized by confocal laser microscopy to analyze whether PIEZO1 is present in human clitoral Krause’s corpuscles and to confirm PIEZO2 expression within them. We were particularly interested in determining whether PIEZO1 and PIEZO2 locations are found in axons, terminal glial cells, or both.

## 2. Results

### 2.1. Identification and Immunohistochemical Profile of Clitoral Krause’s Corpuscles

The first step in the study was to identify Krause’s corpuscles in clitoral sections by immunohistochemistry for specific axonal and glial proteins. Krause’s corpuscles were located at different depths with respect to the clitoris epithelial cover, showed highly variable morphologies and sizes, and occasionally were organized in clusters. The axon was immunoreactive for neuron-specific enolase (NSE) and neurofilament proteins (NFP), but its arrangement and path within the corpuscles were difficult to follow because of a very tightly coiled and “wool ball” or “yarn ball” appearance (Figure 1a). On the other hand, terminal glial cells showed an irregular disposition and displayed an intense immunoreactivity for S100 protein (S100P; Figure 1b). Double immunofluorescence for NSE or NFP and S100P demonstrated intricate relationships between axons and terminal glial cells inside of Krause’s corpuscles (Figure 1c–f) due to immunofluorescence overlapping. The estimated Krause’s corpuscle density from S100P-immunostained sections was 5.22 ± 1.2 per mm^2^, with no evident variations among subjects of different ages.

Throughout the manuscript, the term S100P refers to the use of a polyclonal antibody (which detects the α and β subunits) for the detection of the S100 protein, while S100Pβ refers to the use of a monoclonal antibody (which only detects the β subunit).

### 2.2. Clitoral Krause’s Corpuscles Display Axonal and Glial PIEZO1 Immunoreactivity

When applied, the axons of Krause’s corpuscles were immunolabeled by antibodies against NFP and NSE, while terminal glial cells were immunolabeled by antibodies against S100P. So, double immunofluorescence was used to co-localize PIEZO1 with either axonal or terminal glial cell protein markers. In about 38% of clitoral Krause’s corpuscles, PIEZO1 immunoreactivity was detected apparently restricted to the axon, describing an identical immunostaining pattern for both the axonal NFP or NSE marker ones. The remaining 62% PIEZO1-positive Krause’s corpuscles displayed specific immunoreactivity in both the axons and terminal glial cells, but immunofluorescence was stronger in axon than glial cells (Figure 2). Occasionally, axonal and axonal–glial patterns of PIEZO1 immunofluorescence were detected in different areas of the same corpuscle (Figure 3).

As mentioned before, the terminal glial cells in about 62% of Krause’s corpuscles were immunoreactive for PIEZO1 (Figure 4). Most of the Krause’s corpuscle areas showed merge between the immunofluorescence for PIEZO1 and S100Pβ, meaning that PIEZO1 is localized in glial cells, which showed an irregular and flattened morphology.

### 2.3. Clitoral Krause’s Corpuscles Display Axonal and Glial PIEZO2 Immunoreactivity

PIEZO2 was also detected in Krause’s corpuscles co-localized in most cases only with axonal markers (71; Figure 5a–d). However, immunoreaction for PIEZO2 in 29% of the corpuscles exceeded that of axons, suggesting its presence in terminal glial cells (Figure 5e–h and Figure 6a–c). Analyzing serial sections by immunodetection of PIEZO2 together with NFP or S100Pβ (Figure 4e–h and Figure 6), PIEZO2 was observed to be present in both axons and terminal glial cells.

On the other hand, the comparison of approximate serial sections (40 μm apart) showed a similar pattern of distribution of PIEZO1 and PIEZO2, thus suggesting they co-localize at least in terminal glial cells of Krause’s corpuscles; co-localization in axons cannot be assured (Figure 4).

## 3. Discussion

The skin and cutaneous mucous tissues, like the lip vermillion or the skin-covering glans of both penis and clitoris, are subject to continuous external mechanical forces, and mechanotransduction is of utmost importance to process and leverage mechanical input for its various functions. Therefore, these organs must express mechanosensitive ion channels to fulfill that function. Among them, the most important ones are two members of the Piezo family. PIEZO1 and PIEZO2 are vertebrate multipass transmembrane proteins forming a part of the cationic ion channels, directly involved in mechanotransduction [12,26,27]. In skin, PIEZO1 is widely distributed in the epidermis [25], while PIEZO2 is mainly present in the axon of CEOCs, hair follicle nerves, and Merkel–neurite cell complexes, as well as in some cutaneous non-nervous tissues [12,17,20]. On the other hand, while the critical role of PIEZO2 in touch detection is now well established, it is unknown whether PIEZO1 also contributes to touch sensation. Therefore, if PIEZO1 has any function in mechanotransduction, it will be expressed in some of the components of the sensory corpuscles, especially the axon. Recently, we have demonstrated that the axon of Meissner’s corpuscles in human digital skin display immunoreactivity for PIEZO1 and PIEZO2 [20].

The present study was designed to investigate the presence of PIEZO1 and PIEZO2 in Krause’s corpuscles from human glans clitoridis. Our findings on PIEZO2 expression in the axon are in complete agreement with previous studies in human CEOCs in different anatomical locations [12,18,20,28] but also in mice ones [10,11,12,14,19,29], including clitoral Krause’s corpuscles [9,10]. Furthermore, we observed here that about 30% of these corpuscles display PIEZO2 immunoreactivity in terminal glial cells. As far as we know, this was previously described by Shin et al. [14], who showed the occurrence of PIEZO2 in both the axon and non-neuronal cells of rat CEOCs, and Yamanishi and Iwabuchi [21], who reported PIEZO2 expression in both the axon and terminal glial cells of human lanceolate nerve endings.

Conversely, our positive results obtained for PIEZO1 in Krause’s corpuscles are reported for the first time. Previous studies revealed PIEZO1 immunoreactivity in both the axon of human Meissner’s corpuscles [20] and the axon and terminal glial cells of murine CEOCs [15]. Here, we demonstrate human Krause’s corpuscles display PIEZO1 immunoreactivity in both the axon and the terminal glial cells. PIEZO1 occurrence in sensory corpuscles is consistent with data from PIEZO1-deficient animals, which show deficits in behavioral responses to both light touch and high-threshold mechanical stimuli [30,31].

An unexpected but important result of our study was the demonstration of PIEZO1 and PIEZO2 in terminal glial cells of Krause’s corpuscles. It has been conventionally accepted that the axon of sensory corpuscles is the sole place for transduction of innocuous mechanical stimuli. Nevertheless, this dogma is doubted by recent studies demonstrating how non-neuronal cells, including terminal glial cells, are required for normal detection and coding of somatosensory stimuli in the peripheral nervous system [32,33]. Classically, the terminal glial cells filling CEOCs have been regarded as passive, just mechanical, elements in the process of action potential generation. However, terminal glial cells are revealed to play an important active role in the mechanotransduction process (see [7,34]). Different studies show that a substantial part of the mechanical stimulus transduction into an electrical signal occurs in terminal glial cells, which transfer forces to the axon throughout cell-cell junctions formed between the extensive array of axon protrusions and terminal glial cells [19]. Interestingly, lamellar cells of avian Grandry’s corpuscles (equivalent to mammalian Meissner’s corpuscles) use calcium influx to trigger action potentials in the axon [33]. The mechanotransduction process involves passage of Ca^2+^ through PIEZO channels [16,26]. Thus, the occurrence of PIEZO1 and PIEZO2 in terminal glial cells in Krause’s corpuscles suggest they participate, in addition to the axon, in the mechanosensing and/or mechanotransduction processes.

Krause’s corpuscles are associated with rapidly adapting LTMRs [10] and, because of PIEZO2, are related to fine touch, vibration, motion, and displacement across the skin [35]. The role of PIEZOs in clitoral Krause’s corpuscles remains to be fully elucidated but is presumably related to the involvement of mechanical stimuli in sexual behavior and in sexual pleasure since genital tactile stimulation is a critical component of sexual arousal and orgasmic response [36,37,38,39,40,41]. Nevertheless, it is necessary to consider that the specialized glabrous skin from the glans clitoridis contains combinations of LTMRs that make it functionally distinct and ultimately determine orgasm pleasure sensibility, rendering it as the center for triggering the orgasmic response [24]. Recently, PIEZO2 has been detected in mouse clitoral Krause’s corpuscles [10], whose function is needed for triggering touch-evoked erection reflex and successful mating in both male and female mice. On the other hand, humans with complete loss of PIEZO2 function have genital hyposensitivity and experience no direct pleasure from gentle touch or vibration [24,25,26,27,28,29,30,31,32,33,34,35,36,37,38,39,40,41,42,43,44,45].

Now, the role of PIEZO1 in Krause’s corpuscles is unknown, but its participation alone or in conjunction with PIEZO2 in the process of detecting different modalities of touch and/or vibration may be speculated. Further studies are needed to elucidate the role of PIEZO1 in CEOCs, particularly in Krause’s corpuscles.

## 4. Materials and Methods

### 4.1. Material

The glans clitoridis (n = 6) was obtained from unembalmed female cadavers (age ranged from 46 to 71 years old). Samples were fixed in 4% buffered formaldehyde and routinely processed for paraffin embedding. Subsequently, they were cut into 10 μm thick sections, perpendicular to the surface, and mounted on gelatine-coated microscope slides.

These materials were obtained from the laboratory of the Peripheral Nervous System and Sense Organs Research Group (SINPOs) at the University of Oviedo (Registro Nacional de Biobancos, Sección Colecciones, Ref. C-0001627, Oviedo, Spain). The biological material was obtained in compliance with the Spanish Legislation (RD 1301/2006; Law 14/2007; RD 1716/2011; Order ECC/1404/2013) and in agreement with the guidelines of the Declaration of Helsinki II. This study was approved by the Ethical Committee for Biomedical Research of the Principality of Asturias, Spain (Cod. Celm. Past: Proyecto 266/18).

### 4.2. Simple Immunohistochemistry

Deparaffinized and rehydrated sections were processed for indirect immunohistochemistry using Leica Bond™ Polymer Refine Detection Kit (Leica Biosystems™, Newcastle, UK) following the manufacturer instructions. The primary antibodies used were directed against axonal proteins (NFP: neurofilament proteins; mouse clone 2F11, prediluted, Roche, Vienna, Austria; NSE: neuron-specific enolase; mouse clone BBS/NC/VI-H14; used diluted: 1:200; DAKO, Glostrup, Denmark) and terminal glial cells proteins (S100 protein raised in rabbit; used diluted: 1:1000; DAKO, Glostrup, Denmark). Indirect immunohistochemistry included several negative and positive controls, as well as internal positive and negative controls.

### 4.3. Double Immunofluorescence

The main constituents of Krause’s corpuscles were identified using antibodies against NFP and NSE to immunolabel the axon and against S100 protein (S100P or S100Pβ, mouse clone 4C4.9; used diluted: 1:1000; TermoFisher Scientific, Saint Louis, MO, USA) for terminal glial cells. Sections were deparaffinized, rehydrated, washed in phosphate-buffered saline/Tris (PBS-T), pH 7.4, for 20 min, and antigens were unblocked with PBS + 0.1% Tween. Then, they were incubated overnight at 4 °C in a humid chamber with a 1:1 mixture of two primary antibodies for the simultaneous detection of two antigens: PIEZO1 (raised in rabbit against a synthetic peptide C-EDLKPQHRRHISIR, corresponding to amino acid sequence 1863–1876 of rat PIEZO1; used diluted 1:200; Alomone, Jerusalem, Israel) plus NFP or NSE or S100Pβ. Identical experiments were conducted using PIEZO2 (raised in rabbit against a synthetic peptide VFGFWAFGKHSAAADITSSLSEDQVPGPFLVMVLIQFGTMVVDRALY LRK; used diluted: 1:500; Sigma-Aldrich, Saint Louis, MO, USA). After raising with TBS, sections were incubated with secondary antibodies for 90 min each: first, Alexa Fluor 488-conjugated goat anti-rabbit IgG (1:100; Serotec™; Oxford, UK) and then Cy3-conjugated donkey anti-mouse IgG (1:200; Jackson-ImmunoResearch™; Baltimore, MD, USA). Both steps were performed at room temperature in a dark humid chamber. A PBS-T wash was performed between both incubations. Finally, sections were stained with DAPI (4′,6-diamino-2-phenylindole; 10 ng/mL) to contrast nuclei (blue color) and mounted with diluted Fluoromount-G mounting medium (Southern-Biotech; Birmingham, AL, USA). Immunofluorescence was detected using a Leica DMR-XA automated fluorescence microscope coupled to Leica Confocal Software v2.5 fluorescence capture software (Leica Microsystems, Heidelberg GmbH, Germany) from the Image Processing Service of the University of Oviedo. Specific immunoreaction controls were performed in the same way as for simple immunohistochemistry. Additional controls omitting both antibodies were conducted to confirm absence of tissue autofluorescence or that produced by the fixation process.

### 4.4. Quantitative Study

Ten sections of each glans clitoridis sample, 10 µm thick, 200 µm apart, were processed for S100P immunohistochemistry and used to identify Krause’s corpuscles. Sections were scanned by an SCN400F scanner (Leica Biosystems), and the scans were computerized using SlidePath Gateway LAN software (version 3.1, Leica Biosystems™). Then, in 5 randomly selected fields of 400 µm^2^ each, one per section, the number of sensory corpuscles was counted by two independent observers. Values are expressed as the mean of sensory corpuscles in mm^2^. Due to the low number of sampled corpuscles, no statistical analysis was carried out.

## 5. Conclusions

As a summary, the present study demonstrates that human clitoral Krause’s corpuscles display PIEZO1 and PIEZO2 immunoreactivity in the axon and terminal glial cells. In consequence, both ion channels may be involved in touch. Furthermore, while PIEZO2 seems to mediate specifically light touch originated in sensory corpuscles, PIEZO1 appears to be a more general amplifier of cutaneous mechanical stimuli because of its presence in sensory corpuscles and keratinocytes. The results suggest, in the absence of functional studies, that PIEZO1 and PIEZO2 may participate in the detection of different mechanosensitivity modalities.

## Figures and Tables

**Figure 1 ijms-25-06722-f001:**
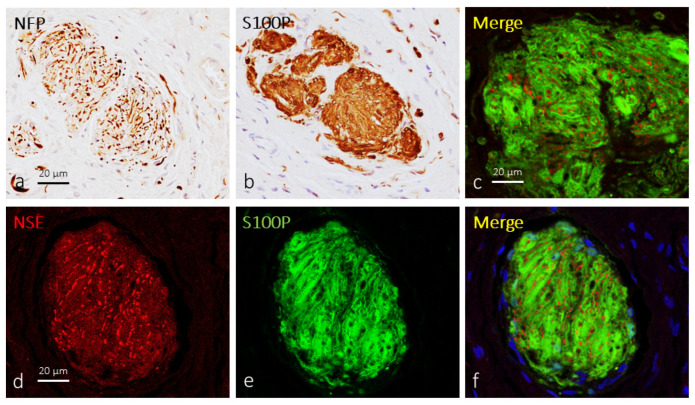
Immunohistochemical profile of human clitoral Krause’s corpuscles. Approximate serial sections of a Krause’s corpuscle (**a**,**b**). Axons were immunoreactive for NFP (**a**) and NSE (**d**) and showed a “wool ball” or “yarn ball” arrangement, while terminal glial cells displayed a strong immunoreactivity for S100P (**b**,**e**). Double immunofluorescence for NSE (red fluorescence in (**c**,**f**)) and S100P (green fluorescence in (**c**,**f**)) showed no merge and the intricate relationships between axons and terminal glial cells in Krause’s corpuscles. Sections processed for double immunofluorescence were counterstained with DAPI (blue) to ascertain structural details. Objective 63×/1.40 oil; pinhole 1.37; XY resolution 139.4 nm and Z resolution 235.8 nm.

**Figure 2 ijms-25-06722-f002:**
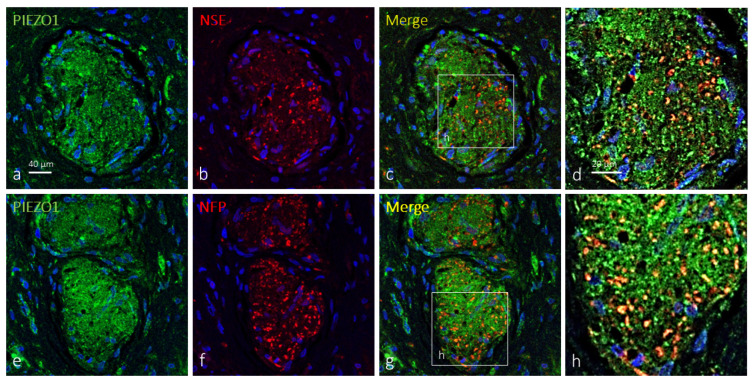
PIEZO1 immunofluorescence in human clitoral Krause’s corpuscles. Double immunofluorescence for PIEZO1 ((**a**,**e**); green fluorescence) and NSE or NFP ((**b**,**f**); red fluorescence). Merge was detected in the axons, but terminal glial cells also displayed PIEZO1 immunostaining. Sections were counterstained with DAPI (blue) to ascertain structural details. (**d**,**h**) are enlargements of the squares in (**c**,**g**), respectively. Objective 63×/1.40 oil; pinhole 1.37; XY resolution 139.4 nm and Z resolution 235.8 nm.

**Figure 3 ijms-25-06722-f003:**
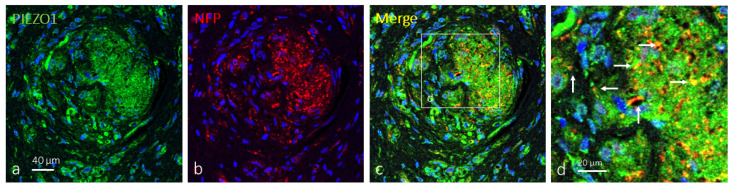
PIEZO1 immunofluorescence in human clitoral Krause’s corpuscles. Double immunofluorescence for PIEZO1 ((**a**); green fluorescence) and NFP ((**b**); axon in red fluorescence). Merge was detected in the axons (arrows), but terminal glial cells also displayed PIEZO1 immunostaining. Sections were counterstained with DAPI (blue) to ascertain structural details. (**d**) is an enlargement of the square in (**c**). Objective 63×/1.40 oil; pinhole 1.37; XY resolution 139.4 nm and Z resolution 235.8 nm.

**Figure 4 ijms-25-06722-f004:**
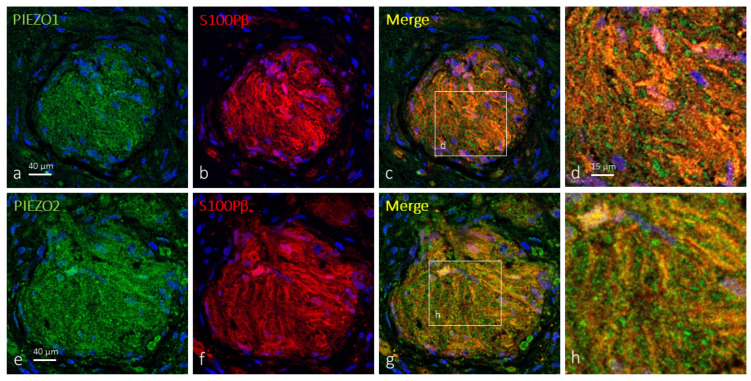
PIEZO1 and PIEZO2 immunofluorescence in approximate serial sections of human clitoral Krause’s corpuscles. Double immunofluorescence for PIEZO1 (**a**) and PIEZO 2 (**e**) (green fluorescence) and S100Pβ (**b**,**f**) (red fluorescence). Merge was detected in terminal glial cells, but axon profiles displaying only PIEZO1 or PIEZO2 immunoreactivity (green fluorescence in points and drops in (**c**,**d,g**,**h**)) were also observed, following the same pattern as axon markers. (**d**,**h**) are enlargements of the squares in (**c**,**g**), respectively. Sections were counterstained with DAPI (blue) to ascertain structural details. Objective 63×/1.40 oil; pinhole 1.37; XY resolution 139.4 nm and Z resolution 235.8 nm.

**Figure 5 ijms-25-06722-f005:**
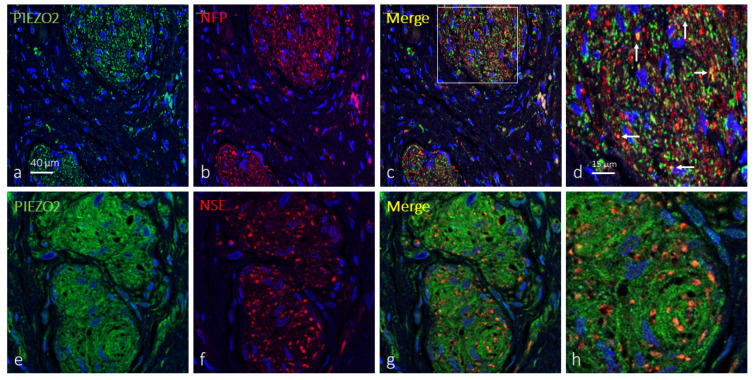
PIEZO2 immunofluorescence in human clitoral Krause’s corpuscles. Double immunofluorescence for PIEZO2 ((**a**,**e**); green fluorescence) and NFP or NSE ((**b**,**f**); red fluorescence). Merge was detected in the axons (see white arrows in **d**), but terminal glial cells also displayed PIEZO2 immunostaining (**g**,**h**). Sections were counterstained with DAPI (blue) to ascertain structural details. (**d**) is an enlargement of the square in (**c**). Objective 63×/1.40 oil; pinhole 1.37; XY resolution 139.4 nm and Z resolution 235.8 nm.

**Figure 6 ijms-25-06722-f006:**
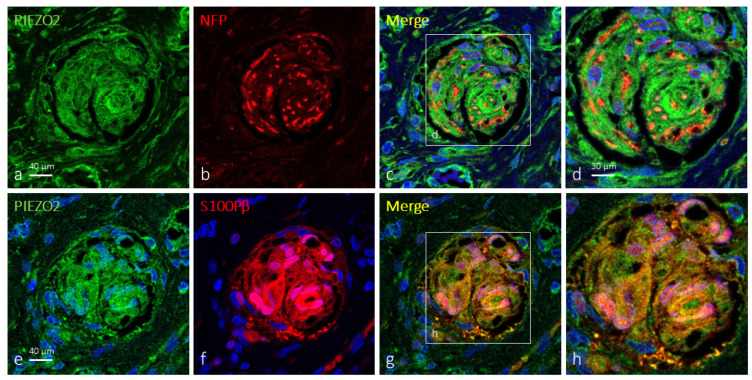
PIEZO2 immunofluorescence in approximate serial sections of human clitoral Krause’s corpuscles. Double immunofluorescence for PIEZO2 (green fluorescence, (**a**,**e**)) and NFP (**b**) or S100Pβ (**f**) (red fluorescence). Merge of PIEZO2 and NFP was detected in the axons (**c,d**) and between PIEZO2 and S100Pβ (**g,h**) in terminal glial cells. Sections were counterstained with DAPI (blue) to ascertain structural details. (**d**,**h**) are enlargements of the squares in (**c**,**g**), respectively. Objetive: 40×/1.40 oil; pinhole 1.37; XY resolution 139.4 nm and Z resolution 235.8 nm (**a**–**c**,**e**–**g**). Objective 63×/1.40 oil; pinhole 1.37; XY resolution 139.4 nm and Z resolution 235.8 nm (**d**,**h**).

## Data Availability

The data that support the findings of this study are available from the corresponding author upon reasonable request.

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
