# Peer review of "Axonal and Glial PIEZO1 and PIEZO2 Immunoreactivity in Human Clitoral Krause’s Corpuscles"

_ijms, 2024, doi:10.3390/ijms25126722_

Round 1
Reviewer 1 Report
Comments and Suggestions for Authors
In the manuscript submitted for review, the Authors analyzed PIEZO1 and PIEZO2 axonal and glial immunoreactivity in human clitoral Krause bodies. I find the subject of the manuscript interesting, especially since, as the Authors mention, there is no information on it. The entire work is perfectly thought out. The Authors put a lot of work into preparing this interesting work. Carefully prepared photos undoubtedly attract the reader's attention.
My comments:
1. Figure 2 and 3: there are squares with letter markings on them, but the caption does not explain what they represent
2. How was the antigen unblocked during double IHC staining? whether high temperature was used: or maybe another method was used; there is no such information
3. Because the study is innovative, did the Authors think about using, for example, Western Blot? It would certainly enrich the manuscript
4. Did the Authors observe any differences in the results between the clitoris from the cadavers of a younger and an older woman, the difference was about 30 years;
Author Response
REVIEWER 1
The authors thank the anonymous Reviewer for the constructive criticism. Thank you very much. In agreement with the referees' suggestions, the following changes have been made in the manuscript (labelled in red):
In the manuscript submitted for review, the Authors analyzed PIEZO1 and PIEZO2 axonal and glial immunoreactivity in human clitoral Krause bodies. I find the subject of the manuscript interesting, especially since, as the Authors mention, there is no information on it. The entire work is perfectly thought out. The Authors put a lot of work into preparing this interesting work. Carefully prepared photos undoubtedly attract the reader's attention.
My comments:
- Figure 2 and 3: there are squares with letter markings on them, but the caption does not explain what they represent.
Thanks for the comment. The squares, and the letter within them, frame part of the figures shown at higher magnifications. The legend of the figures has been modified to clarify this aspect.
- How was the antigen unblocked during double IHC staining? whether high temperature was used: or maybe another method was used; there is no such information.
To unblock antigens during double IHC, PBS with 0.1% Tween was used for 15 minutes. This technical comment was added to the revised version.
- Because the study is innovative, did the Authors think about using, for example, Western Blot? It would certainly enrich the manuscript.
The suggestion is very important in order to characterize the immune-labeled protein. Unfortunately, at the present time we do not have fresh material to be able to make it. On the other hand, as we use material from fresh human cadavers, for legal reasons, the material cannot be obtained in the immediate postmortem and this makes the available material not suitable for Westen Blot studies. We plan to carry out molecular biology studies on material from multi-organ donors perfused with proteolytic enzyme blockers, but we do not currently have this material.
- Did the Authors observe any differences in the results between the clitoris from the cadavers of a younger and an older woman, the difference was about 30 years.
In the present work, we have not observed variations in the density of Krause corpuscles in the age range studied. In a previous study (Y. García-Mesa et al. J Anat. 2021; 238(2):446-454. doi: 10.1111/JOA.13317) a non-significant age-dependent reduction in Krause corpuscle density was observed, a fact that we have not observed in the present study, perhaps due to the small sample size.
Reviewer 2 Report
Comments and Suggestions for Authors
In this manuscript, Patricia Cuendias et. al., demonstrate for the first time that PIEZO1 is expressed in terminal glial cells of Kraus's corpuscles. Authors have used human female glans clitoris between age of 46-71 years old, and performed immnunohistochemetry. Understanding the functions of PIEZOs is important for sensations and touch. For better understating and improving the quality of manuscript, I have minor concerns below-
1. Write the S100Pβ consistently throughout the manuscript. At some place, its written S100P, and someplace S100Pβ.
2. Increase the font size of text written inside the images or bold them for better visibility. This is true for all images incorporated in the manuscript.
3. In figure 2 and 3, authors descibed in the result "The remaining 62 ± 5% PIEZO1 positive Krause’s corpuscles displayed specific immunoreactivity in both the axons and 104 terminal glial cells, but immunofluorescence was stronger in axon than glial cells (Figs. 2 and 3). " that PIEZO1 is expressed in terminal glia. However, S100Pβ image is not in the figure 2 or 3. Might be author has mislabelled with NSE, if so please correct that. Additionally, Figure 2 and 3 images are describing the same result. It would be important to quantify the co-localization of two proteins, rather than showing multiple images for the result. This is true for the rest of the figures such as Figures 4 & 5; as well as Figures 6 & 7.
Author Response
REVIEWER 2
The authors thank the anonymous Reviewer for the constructive criticism. Thank you very much. In agreement with the referees' suggestions, the following changes have been made in the manuscript (labelled in red):
In this manuscript, Patricia Cuendias et. al., demonstrate for the first time that PIEZO1 is expressed in terminal glial cells of Kraus's corpuscles. Authors have used human female glans clitoris between age of 46-71 years old and performed immnunohistochemistry. Understanding the functions of PIEZOs is important for sensations and touch. For better understating and the quality of manuscript, I have minor concerns below-
1.Write the S100Pβ consistently throughout the manuscript. At some place, its written S100P, and someplace S100Pβ.
Thanks for the suggestion. Throughout the manuscript, both S100 and S100β are used in two different contexts: S100P when it is a polyclonal antibody (as it reacts with both the α and β subunits, and S100β when it is a monoclonal antibody that recognizes only that subunit). A brief senencia has been introduced in the manuscript to clarify the use of double terminology. Image captions have been modified accordingly.
- Increase the font size of text written inside the images or bold them for better visibility. This is true for all images incorporated in the manuscript.
Thanks for the observation. The font size has been increased (from 16 to 20) and the immunoreactivity identifiers have been transformed into bold.
- In figure 2 and 3, authors described in the result "The remaining 62 ± 5% PIEZO1 positive Krause’s corpuscles displayed specific immunoreactivity in both the axons and 104 terminal glial cells, but immunofluorescence was stronger in axon than glial cells (Figs. 2 and 3). " that PIEZO1 is expressed in terminal glia. However, S100Pβ image is not in the figure 2 or 3. Might be author has mislabelled with NSE, if so please correct that.
Additionally, Figure 2 and 3 images are describing the same result. It would be important to quantify the co-localization of two proteins, rather than showing multiple images for the result. This is true for the rest of the figures such as Figures 4 & 5; as well as Figures 6 & 7.
Thank you very much for the detailed remarks. As can be seen, the figures have been reorganized to avoid redundancies and make the locations of the investigated antigens clearer. The images of serial sections have been maintained to observe that PIEZO1 and PIEZO2 are in the same locations, i.e., that PIEZO1 and PIEZO2 can be co-located in the same cells.
In terms of quantification, it has not been possible to quantify the area occupied by the localization for PIEZO1 and PIEZO2 because there is merging in very large areas that the gray detection (false color) softward could not differentiate. Obviously, the quantification of proteins as a function of the intensity of the immunoreaction is not adequate because it is well known that there is no stoichiometric relationship between the amount of protein and the intensity of the immunoreaction.
Reviewer 3 Report
Comments and Suggestions for Authors
The authors demonstrate the presence of PIEZOs 1 & 2 in the axon terminals and accessory (glial) cells in Krause’s corpuscles of the glans clitoridis, using immunohistochemistry and immunofluorescence. The results for PIEZO2 are confirmatory, whereas those for PIEZO1 are new.
There are several minor problems with English, most of which I leave to the editorial team to correct.
I have a few, relatively minor comments:
L 20 and elsewhere: “gland clitoris” is not a recognised anatomical term – clitoral glans, glans of the clitoris, or (preferably, perhaps) glans clitoridis would be better.
L 25 and elsewhere “woman” is not adjectival in English. “Human” would be better.
L 49 I doubt that any single type of sense organ could be said to detect “sexual pleasure”.
L 101-106 (and elsewhere) are the mean limits (e.g. 38 ± 8%) the standard deviation? These two sentences read as though PIEZO1 was not detected in terminal glial cells in about 38 % of corpuscles. Is this correct?
L 103-106 “The remaining 62 ± 5% PIEZO1-positive Krause’s corpuscles displayed specific immunoreactivity in both the axons and terminal glial cells, but immunofluorescence was stronger in axon than glial cells (Figs. 2 and 3)." I have looked very carefully at figs 2 and 3, but as far as I can see the IMF was generally stronger in the glial cells in those figures.
L 152-153 “PIEZO2 was also detected in Krause’s corpuscles co-localized in most cases only with axonal markers (71 ± 8%; Fig. 6).” Figure 6 shows PIEZO2 immunofluorescence in both axons and glial cells. If, as the sentence states, only the axons were labelled in most corpuscles, it would be useful to show an example.

Nothing to add to the above.
Author Response
REVIEWER 3
The authors thank the anonymous Reviewer for the constructive criticism. Thank you very much. In agreement with the referees' suggestions, the following changes have been made in the manuscript (labelled in red):
The authors demonstrate the presence of PIEZOs 1 & 2 in the axon terminals and accessory (glial) cells in Krause’s corpuscles of the glans clitoridis, using immunohistochemistry and immunofluorescence. The results for PIEZO2 are confirmatory, whereas those for PIEZO1 are new.
There are several minor problems with English, most of which I leave to the editorial team to correct.
I have a few, relatively minor comments:
L 20 and elsewhere: “gland clitoris” is not a recognized anatomical term – clitoral glans, glans of the clitoris, or (preferably, perhaps) glans clitoridis would be better.
Thanks for the suggestion. The substitution has been made throughout the manuscript including the Abstract.
L 25 and elsewhere “woman” is not adjectival in English. “Human” would be better.
Done.
L 49 I doubt that any single type of sense organ could be said to detect “sexual pleasure”.
It has been replaced by ‘sexual mechanical sensing’
L 101-106 (and elsewhere) are the mean limits (e.g. 38 ± 8%) the standard deviation? These two sentences read as though PIEZO1 was not detected in terminal glial cells in about 38% of corpuscles. Is this correct?
The quantitative analysis has been thoroughly reviewed and since the results of Krause corpuscles in which axonal or glial expression predominates, or both are expressed in percentages, the results have been adjusted to whole numbers. We consider that in this way the results are less confusing.
L 103-106 “The remaining 62 ± 5% PIEZO1-positive Krause’s corpuscles displayed specific immunoreactivity in both the axons and terminal glial cells, but immunofluorescence was stronger in axon than glial cells (Figs. 2 and 3)." I have looked very carefully at figs 2 and 3, but as far as I can see the IMF was generally stronger in the glial cells in those figures.
This question was already answered partially in the previous paragraph. Admittedly, it may be difficult to interpret, but without any doubt the "droplet" pattern of axon profiles has a higher fluorescence intensity than that of glial end cells, which is diffuse. In addition, the distribution pattern of PIEZO1 and PIEZO2 considered of axon profiles merges with that of axon markers. In the figures modified in this revision, the contrast has been increased to better see the distribution pattern.
L 152-153 “PIEZO2 was also detected in Krause’s corpuscles co-localized in most cases only with axonal markers (71 ± 8%; Fig. 6).” Figure 6 shows PIEZO2 immunofluorescence in both axons and glial cells. If, as the sentence states, only the axons were labelled in most corpuscles, it would be useful to show an example.
Thanks for the suggestion. A figure (5a-d) has been included to illustrate this information.